# Prediction Model for Identifying Factors Associated with Epilepsy in Children with Cerebral Palsy

**DOI:** 10.3390/children9121918

**Published:** 2022-12-08

**Authors:** Carlo Mario Bertoncelli, Nathalie Dehan, Domenico Bertoncelli, Sikha Bagui, Subhash C. Bagui, Stefania Costantini, Federico Solla

**Affiliations:** 1Department of Computer Science, Hal Marcus College of Science & Engineering, University of West Florida, Pensacola, FL 32514, USA; 2EEAP H Germain & Department of Pediatric Orthopaedic Surgery, Lenval University Pediatric Hospital of Nice, 06200 Nice, France; 3Department of Information Engineering Computer Science and Mathematics, University of L’Aquila, 67100 L’Aquila, Italy; 4Lenval University Pediatric Hospital of Nice, 06200 Nice, France

**Keywords:** cerebral palsy, epilepsy, statistics, machine learning

## Abstract

(1) Background: Cerebral palsy (CP) is associated with a higher incidence of epileptic seizures. This study uses a prediction model to identify the factors associated with epilepsy in children with CP. (2) Methods: This is a retrospective longitudinal study of the clinical characteristics of 102 children with CP. In the study, there were 58 males and 44 females, 65 inpatients and 37 outpatients, 72 had epilepsy, and 22 had intractable epilepsy. The mean age was 16.6 ± 1.2 years, and the age range for this study was 12–18 years. Data were collected on the CP etiology, diagnosis, type of epilepsy and spasticity, clinical history, communication abilities, behaviors, intellectual disability, motor function, and feeding abilities from 2005 to 2020. A prediction model, Epi-PredictMed, was implemented to forecast the factors associated with epilepsy. We used the guidelines of “Transparent Reporting of a multivariable prediction model for Individual Prognosis or Diagnosis” (TRIPOD). (3) Results: CP etiology [(prenatal > perinatal > postnatal causes) *p* = 0.036], scoliosis (*p* = 0.048), communication (*p* = 0.018), feeding disorders (*p* = 0.002), poor motor function (*p* < 0.001), intellectual disabilities (*p* = 0.007), and the type of spasticity [(quadriplegia/triplegia > diplegia > hemiplegia), *p* = 0.002)] were associated with having epilepsy. The model scored an average of 82% for accuracy, sensitivity, and specificity. (4) Conclusion: Prenatal CP etiology, spasticity, scoliosis, severe intellectual disabilities, poor motor skills, and communication and feeding disorders were associated with epilepsy in children with CP. To implement preventive and/or management measures, caregivers and families of children with CP and epilepsy should be aware of the likelihood that these children will develop these conditions.

## 1. Introduction

Cerebral palsy (CP) encompasses nonprogressive motor and postural control disorders that arise as a result of brain damage during early development [1]. Comorbidities in children with CP constitute a significant problem in this population. CP is also associated with a higher incidence of seizures. The most frequent seizure types are complex focal and secondary generalized seizures, for which an early evaluation is strongly recommended [2] as children with CP tend to have an earlier epilepsy onset, and the degree of severity is positively correlated with the CPs severity.

According to the etiology and topography of the cerebral lesions, speech difficulties, hearing and visual impairments, and intellectual disabilities may be associated with epilepsy [1].

Children with CP often present autistic features marked by communication, interaction, and adaptation difficulties. Intellectual disabilities constitute a unique difficulty for epilepsy in children with CP. Typically, these children cannot describe epileptic events, parents describe them with apprehension, and epilepsy specialists rarely witness them [2]. The clinical course is not well defined, although epilepsy occurs in up to 90% of children with CP [1].

Many molecular pathways are implicated in the apoptosis of the premyelinating oligodendrocytes or subplate neurons involved in perinatal brain development. Glutamate rising concentrations or free radical reactive species (both oxygen and hydrogen) in hypoxic-ischemic encephalopathy, inflammatory cytokines such as TNF-α, IL-1b, IL-6, 12, 15, and 18 from activated microglia and astrocytes, and a low pH in infections, free iron secondary to a cerebral hemorrhage are widely reported in both white and grey matter lesions as important triggers for epileptic events [3].

The assessment of children with CP with various comorbidities is difficult. Previous research involved small [4] or limited [5] sample sizes, and there is no information on the accuracy of the predictions [6,7]. There are few studies evaluating related issues despite epilepsy being common in children with CP [7]; there is no accurate data on the factors associated with epilepsy in children with CP using a prediction model. A machine-learning model, PredictMed, has been implemented and validated to predict, in children with CP, the onset of neuromuscular scoliosis and feeding disorders requiring gastrostomy [8], the factors associated with intellectual disabilities [9], autism spectrum disorder [10], and neuromuscular hip dysplasia [11]. Thus, we aimed to adapt and test Epi-PredictMed to identify the factors associated with epilepsy in children with CP.

## 2. Materials and Methods

### 2.1. Design

A multicenter retrospective longitudinal study.

### 2.2. Participants

Seven hundred and seventy-five children with CP were selected according to the following inclusion criteria:-Age: 12–18 years.-With spasticity and/or dystony and/or hypotony according to the Surveillance of Cerebral Palsy in Europe [12].-Minimum follow-up of 3 years.

The exclusion criteria were:-Progressive encephalopathy or spinal disorder.

One hundred and two children (65 hospitalized, 37 in day hospital) met the inclusion criteria, as follows:-A division of 58 males and 44 females.-In total, 62% were white, 32% were Arab, 4% were black, and 2% were Asian.-The age ranged from 12 to 18 years, average 17 ± 1.-The follow-up time ranged from 3 to 12 years, average 6 ± 1.

Pediatric neurologists ascertained the presence and severity of the epilepsy. Before the age of two, a child psychologist made the initial diagnosis of intellectual disability (ID) and interaction and adapting functioning (IAF).

### 2.3. Measures and Procedures

A multidisciplinary healthcare team, including neuro-pediatricians, orthopedic surgeons, physical therapists, child psychologists, and epidemiologists collected data from 2005 to 2020 from medical records. To minimize the biases regarding the diversified academic background of the researchers, only members working together for at least 10 years were included in the study and the results were discussed periodically under the supervision of a child epileptic senior specialist.

Narrative notes on the CP diagnosis, etiology, topography of the spasticity, epilepsy, functional assessments, and medical history were coded and entered into an electronic database [8]. The data verification and collection for the development of the Epi-PredictMed model has been performed from June to December 2017, while the data analysis began at the end of 2018 and lasted for 2 years.

The distribution of patients according to the functional assessments used are shown in Figure 1:

CP etiology was classified as:Prenatal (genetic, cerebral malformation, infectious, or vascular)Perinatal (anoxic, ischemic, or infectious)Postnatal (traumatic, infectious, epilepsy, or postnatal anoxic/ischemic injury) [8].

A child psychologist assessed the severity of the intellectual disability using the DSM-5 after 2013, the DSM-4 before 2013 (American Psychiatric Association 2013), and the Wechsler Intelligence Scale for Children (WISC) [9]. Intellectual disability has been defined as “mild,” “moderate,” and “severe” depending on the level of support required [9], rather than the IQ scores. He also determined the presence of autism spectrum disorders or autistic features. Autism spectrum disorders were diagnosed if children had one of the diagnoses listed under the F84 category of the International Classification of Diseases (ICD-10) [10]. The neuropsychiatrist reported a long-term psychotropic treatment.

The neurologic status was assessed based on the severity of the deficit, i.e., hemiplegia, diplegia, tri/quadriplegia, the type of tonus disorder (hypertonia or dystonia), and the severity of the epilepsy. The motor deficit was determined through the modified Ashworth Scale of Bohannon and Smith and the modified Tardieu Scale [8].

Pediatric neurologists have determined the presence of epilepsy, according to the International League Against Epilepsy, if there were ≥2 afebrile seizures occurring beyond the neonatal period [13]. The severity of the epilepsy was classified as “intractable” in case of a persistence of seizures despite an adequate administration of at least two appropriate antiepileptic agents [14]. Otherwise, the epilepsy was considered “well controlled”.

It has been considered active epilepsy when two or more unprovoked seizures have occurred during the previous year [13]. Seizures in neonatal patients were detected by paroxysmal EEG changes and were classified by the same descriptors as other seizures. No further classification of the type of epilepsy was made due to the difficulty of describing the seizures by the patients and their parents [14]. In our cohort, there was no other therapeutical interventions such as a ketogenic diet [15].

Scoliosis was determined by a Cobb angle >10° on the spine radiograph and was considered “severe” if the Cobb angle exceeded 40° [8].

### 2.4. Statistical Analysis and Prediction Model

For the development and validation of Epi-PredictMed, we followed the guidelines of the “Transparent Reporting of a multivariable prediction model for Individual Prognosis or Diagnosis” (TRIPOD) [16]. The patient’s data, including their demographic information, functional diagnoses, neurological, and cognitive assessments, were treated anonymously.

Our cohort was separated into two groups (with and without epilepsy) to generate the predictive model during the first stage. The potentially predictive factors ranked were:-Etiology of CP (ET).-Level of intellectual disability (ID).-Type of spasticity (SP).-Presence of scoliosis (NS).-Feeding ability (F).-Gender (SE).-GMFCS.-EDACS.-CFCS (C).-Autistic features (A).

We employed Fisher’s exact tests [6] to identify the associations, confidence intervals, and distributions. We then used OpenEpi software 3.01, an epidemiological application, and Med-Calc^®^ statistical software 20.123 [8] to calculate the odds ratios (OR) (logarithmic and linear), 95% confidence intervals with the referenced *p*-value of 0.05, and z-statistics [8].

During the second stage, the model produced all the possible combinations (tuples) of the 10 predictive factors evaluated, for a total of 511 combinations/tuples (each tuple had between one and nine variables).

In the third stage, logistic regression was performed for each tuple, and its predictive performance was tested. The significant variables with a relaxed value of *p* < 0.2 [17,18] were used as the input variables in a multiregression model using the open-source software R 4.2.2 [8]. The glm() function was used to predict each patient’s probability of epilepsy.

Based on the statistical learning pathway as proposed by Vapnik [8], we ranked the patients’ data into a training set and a test set. We trained the logistic regression on a training set of 82 patients. The trained model allowed for predicting the probability of epilepsy for the 20 patients in the test set. This purpose used ten predictive factors: ID, A, ET, SP, SE, GMFCS, F, NS, EDACS, and C. For instance, facing the 6-element tuple, ID + A + SP + ET + SE + GMFCS, we used the data from all 102 patients divided into a training set and a test set to implement a logistic regression model that could predict epilepsy. We applied a cross-validation by randomly producing 20 different couples of training and test sets to minimize the dependence on the compositions of the training and test sets. The compositions of the training and test sets were randomly varied in 20 rounds of cross-validation.

For each couple, we measured the performance in terms of the accuracy, sensitivity, and specificity of the predictions and averaged the three. For this, we first calculated the rates of the true positives (TP), true negatives (TN), false negatives (FN), and false positives (FP). The sensitivity is the percentage of subjects identified as such (TP/(TP + FN)). The specificity is the percentage of children without CP identified as such (TN/(TN + FP)). The accuracy is the percentage of TP and TN in all assessments—the correct classification: (TP + TN)/(TP + TN + FP + FN) [8]. 

The prediction performance was evaluated on a test data set using the R predict function glm () (a). For each patient in the test set, it produced a probability (Prob). For example, in the case of a tuple of 8 elements:A + ET + SE + NS + CFCS + GMFCS + ID + SP

It would have the form:P (E = yes | glm (A + ET + SE + NS + CFCS + GMFCS + ID + SP))
having, 0 < Prob < 1.

We defined the threshold of the decision boundary. If, for example, P (E = yes | glm (:A + ET + SE + NS + CFCS + GMFCS + ID + SP)) > threshold, then Epi-PredictMed correctly predicted the presence of the epilepsy. We tested the thresholds from 0.1 to 0.8 and compared the results’ accuracy, sensitivity, and specificity [18]. For example, according to the prediction model, choosing 0.3 as the threshold, patients with a probability >0.3 were classified as probably epileptic. Patients with epilepsy were classified as TP (true positive) and without FP (false positive). Similarly, we also assessed the TN (true negative) or FN (false negative). Once the predictions were completed for each patient in the test set, we compared the predictions with the known status of the patients (epileptic “YES” or “NO”) to estimate the accuracy, sensitivity, and specificity of the logistic regression algorithm. 

We had no missing data.

We identified the tuple (among 511) in the fourth stage with the best predictive performance.

Finally, we exploited the tuple found in stage 4 to predict the presence of epilepsy in new patients. We used feature reduction to eliminate the redundancy among the predictive factors and maximize the predictive performance. We reduced the number of variables used in the logistic regression from ten to eight.

## 3. Results

### 3.1. Sample

Most participants (62%) had prenatal CP; 26% had perinatal CP and 12% had postnatal CP. Table 1 shows the description of the clinical data. Epilepsy was observed in 70% of the participants; 32% had intractable epilepsy and 64% had epilepsy at age < 1 year (54% at birth). Most subjects (61%) had a profound intellectual disability (ID), 26% had a severe ID, and 13% had a moderate ID. Autistic features were observed in 27% of the participants, and 68% had epilepsy. Almost a quarter (27%) were taking long-term psychotropic treatments (antipsychotics 66%, antidepressants 34%). The most frequent neurological deficit with spasticity was tri/quadriplegia (50%), followed by lower limb diplegia (15%) and hemiplegia (10%). Epileptic children with CP had a greater risk of scoliosis; 40 participants (39%) had scoliosis, of which 19% had controlled epilepsy, and 15% had intractable epilepsy. The motor skills, communication abilities, ID levels, and eating and drinking capacities are summarized in Figure 1.

Fisher’s exact tests showed the following variables associated with epilepsy (Table 2):Spasticity (OR = 7.1, *p* < 0.001).Neonatal seizures (OR = infinity, *p* < 0.001).Communication disorders with CFCS score 5 (OR = 2.6, *p* = 0.049).Scoliosis (OR = 3.5, *p* = 0.014).Severe intellectual disability (OR = 3.26, *p* < 0.001).Feeding disorders (OR = 3.9, *p* = 0.018) in patients with EDACS score 5 (OR = 3.5, *p* = 0.032).With gastrostomy placement (OR = 4.8, *p* = 0.014).Taking neuroleptics (OR = 0.18, *p* < 0.001).Truncal tone disorders (OR = 3.9, *p* = 0.004).History of surgery (OR = 2.8, *p* = 0.030).Walking disabilities (OR = 4.7, *p* < 0.001).SIS MED score >11 (OR = 7.9, *p* = 0.003).SIS BEHEV > 8 (OR = 2.6, *p* = 0.049).Poor manual abilities with MACS score 5 (OR = 15, *p* < 0.001).Poor gross motor function with GMFCS 5 (OR = 5.1, *p* < 0.001).

Multivariate analysis showed the following factors significantly linked with epilepsy (Table 3):CP etiology [(prenatal > perinatal > postnatal causes) OR = 2.46, SE = 0.43, *p* = 0.036].Scoliosis (OR = 2.96, SE = 0.55, *p* = 0.048).High CFCS score (OR = 2.19, SE = 0.33, *p* = 0.018).GMFCS score (OR = 1.97, SE = 0,41, *p* < 0.001).EDACS score (OR = 1.65, SE = 0.17, *p* = 0.002).Profound intellectual disability (OR = 2.55, SE = 0.35, *p* = 0.007).Neurological deficit associated with spasticity [(quadriplegia/triplegia > diplegia > hemiplegia) OR = 1.86, SE = 0.20, *p* = 0.002].

### 3.2. Logistic Regression

The increasing of ET (postnatal > perinatal > prenatal causes), NS, CFCS, GMFCS, ID, SP (Quadriplegia/triplegia > Diplegia > hemiplegia), and EDACS are associated with epilepsy (in the “Odds Ratio-Linear” column), meaning that for each unit increase in GMFCS, the log odds = ln(*p*/1 − *p*) increases 2.54-fold (where *p* = probability of having epilepsy). The column “Prob (>|z|)” shows the strength of significance of the respective parameter in terms of the *p*-value as a predictor of the presence of epilepsy. This means that the significance of ET, NS, CFCS, GMFCS, ID, SP, and EDACS in predicting the presence of epilepsy is highly probable, with a *p*-value < 0.05.

The best regression model score had an accuracy of 74%, a sensitivity of 98%, a specificity of 73%, and a 82% average score

## 4. Discussion

Epilepsy is frequent in children with CP, but few studies are available [2,7]. In addition, a prediction model has never been used to assess the prevalence of epilepsy in children with CP. Machine learning has been primarily used in biomedical research for tumor classification, new drug discovery, genomics, and the interpretation of diagnostic images [19]. To our knowledge, Epi-PredictMed is the first predictive machine-learning model implemented to identify the variates associated with epilepsy.

Previous studies [8] stated that CP children with epilepsy were twice as likely to develop scoliosis and truncal tone disorders and undergo surgeries. Epi-PredictMed scored an average accuracy, sensitivity, and specificity of 82%, improving the previous results of 74% [8].

Epilepsy could therefore be a comorbidity of a deficient nervous system with musculoskeletal disorders; encephalopathies inducing epilepsy commonly impact motor control. This could produce postural or truncal tone disorders. An early cortical lesion can explain the association between truncal tone disorder and epilepsy [8]. There is little literature on this, probably due to the difficulty in combining the patient data relating to epilepsy with orthopedic data. As previously reported [20,21], we also confirm the link between motor deficiency and the extent of spasticity with epilepsy, underlining the two-time ratios. We have partially confirmed Archana et al.’s recent study [22], which dealt with the subtypes of spasticity (spastic hemiplegia > quadriplegia > diplegia > mixed type CP > dyskinetic CP) and the predominance of poor Gross Motor Function Classification System (GMFCS) scores (>III) in epileptic children with CP. In the present study, hemiplegia is the least correlated to epilepsy. This is probably due to the different compositions of the cohorts.

The present study stated that they are likely (odds ratio > 2) to show feeding disabilities, needing gastrostomy. As highlighted by Dahlseng et al. [23], motor deficiency, the extent of spasticity, feeding disabilities, namely needing gastrostomy, and epilepsy are strongly related. Artificial feeding should be proposed after a multidisciplinary concertation to avoid deficiencies (vitamins and trace elements) that facilitate the occurrence of epileptic seizures in an epileptic patient. Dysphagia or digestive troubles must be considered in an epileptic patient who must swallow antiepileptic drugs regularly every day. Additionally, the decision to feed a child with a gastrostomy needs to be made appropriately after a multidisciplinary assessment. Caregivers and parents have to be included; considering acceptance also depends on cultural parameters.

We also detected a relevant association (odds ratio > 2) between epilepsy and profound intellectual and communication disabilities, which confirms previous reports [20,21,24]. In addition, in the present study, the Epi-Predicted sensitivity of 98% allows for the more precise identification of children at risk of epilepsy. Additionally, this allows for the personalization of treatments.

In addition, in these cases, the supervisory and control role of the central nervous system is evident, and its malfunction can generate multiform disorders. This could partly explain this population’s limited social and professional insertion; epilepsy can limit a proper insertion in affected young adults [25,26]. The observation that epilepsy was associated with an intellectual disability is relevant because it associates epilepsy and intellectual disability with grey matter dysfunction [9].

We also found a link between the type of etiology of CP (prenatal > perinatal > postnatal causes) and epilepsy with a double odds ratio. The maturing nervous system is more likely to be affected by epilepsy. Neonatal seizures during the first year of life were found to be related to a significantly increased risk of epilepsy in children with CP, as previously reported [7,27]. In the context of CP, epilepsy may have a better prognosis in older children [28].

All these comorbidities are significant to detect and treat. There has to be an improvement in the prognosis of children with CP. In cases where children with CP have epilepsy with the following factors, we propose specific early interventions:Maintaining an upright posture and strengthening the back muscles through walking appear to protect against severe scoliosis during growth. To assess the onset of scoliosis early and identify possible candidates for treatment, the frequency of clinical spine examinations and spinal X-rays should be increased in cases of a high risk of the onset of scoliosis [8].Postural disorders (due to scoliosis and truncal tone disorders) may require specific physical therapy and/or orthopedic and/or surgical treatment.A high risk of falls due to spasticity, lower limbs paresis or plegia, and motor deficiency could lead to complications linked to seizures; therefore, specific orthoses must be provided.Feeding disorders with eventual restrictive respiratory insufficiency related to severe scoliosis could lead to respiratory complications in cases of a loss of consciousness associated with seizures.

These children need more careful supervision. Parents should be advised early in the process of the probable need for a gastrostomy; this will enable them to understand the benefits better and accept this invasive procedure. Consequently, when gastrostomy becomes necessary, parents will not be slow to accept the medical decision. Comorbidities can be limited by providing for malnutrition. An early dietary follow-up can be set up and will allow for checking if food intake is adequate. Performing regular anthropometric assessments is very important. Even with the challenges in the evaluation to obtain simple measures like weight and height in some children with CP, there are alternative methods to assess nutritional status. A toolkit was recently created to help caregivers to detect malnutrition in children with cerebral palsy [29].

An intellectual assessment and educational guidance are critical to properly advise patients and families of study options [30].Since the type of the etiology and neonatal seizures are predictors of epilepsy, one could anticipate the treatments, given the changes (physical, cerebral, social, and professional) that must be considered in the particular setting [31,32].

The main implications of our findings include:

Warning the caregivers and families of epileptic CP children about the risks of developing the above features.Helping clinicians with decision-making by starting any antiepileptic drug. The choice of the molecule remains a multidimensional reflection, including the assessment of the young patient’s comorbidities, age, and way of life [33,34,35]. Every young patient with CP, having new focal neurological signs, a recent abnormal movement, or acute consciousness modification should have a pediatric or neurologic consultation and electroencephalogram as quickly as possible.Having an early orientation regarding the non-invasive techniques of cerebral activity assessment, such as the electroencephalogram. This can quickly provide the necessary data in the context of presumed seizures [36]. This exam may show sharp waves or abnormalities in the presumed interictal period (especially in some encephalopathies linked to CP) that need trained EEG readers [37].The accurate identification of the risk factors (such as in the present study) can facilitate epidemiological research and health planning in population-based disease studies [38]. However, future studies are needed to validate the diagnosis of CP disorders within administrative databases [39].

An algorithm that includes these parameters could help clinicians anticipate multiple diagnoses and make the most appropriate treatment decision.

The best machine-learning model score obtained was an accuracy of 74%, a sensitivity of 98%, a specificity of 73%, and an average score of 82%. This particularly high sensitivity could allow clinicians to (a) rule out an inadequate diagnosis of epilepsy in case of the absence of obvious signs or (b) predict it during a medical follow-up with caregivers.

The model’s validity is confirmed by the fact that the other parameters that the model found (spasticity, intellectual, and motor disability) align with and match the current literature [9,20,21,22,23,24,25,26].

The strengths of this study are the multicenter evaluation, the cross-validation analysis, and the absence of missing data. Our next goal is to modify the model to perform a multiclass classification. A multiclass classification will allow us to handle dependent variables with more than two values, such as different types of epilepsy. We also plan to implement the PredictMed model into integrated clinical decision support systems [40].

The main limitations of our study were the relatively small number of patients included that can lead to an approximate evaluation of the predictive performance of our algorithm and the retrospective nature of the study with the possibility of residual confounding. Therefore, the Epi-PredictMed model’s performance should be interpreted with caution. Therefore, we plan a multicentric research study employing randomized control trials on larger cohorts to address these issues.

## 5. Conclusions

We implemented a prediction model that can identify the factors associated with epilepsy in children with CP. CP etiology (prenatal), communication and feeding disorders, spasticity, scoliosis, severe intellectual disability, and poor motor skills were associated with epilepsy in children with CP. The model scored 98% on sensitivity, 73% on specificity, and 74% on accuracy (82% on average).

## Figures and Tables

**Figure 1 children-09-01918-f001:**
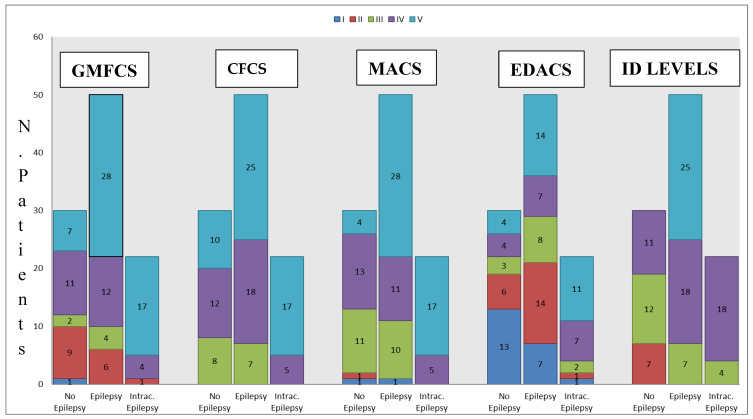
Distribution of patients according to the Manual Ability Classification System [MACS], Gross Motor Function Classification System (GMFCS), Eating and Drinking Ability Classification System (EDACS), Communication Function Classification System (CFCS), and Intellectual Disability (ID) levels [6].

**Table 1 children-09-01918-t001:** Characteristics of participants with the presence and type of epilepsy.

Patient Characteristics	No Epilepsy	Epilepsy	Intractable Epilepsy	Total
Patients, *n*. (%)	30 (29)	50 (49)	22 (22)	102 (100)
Male	18 (18)	26 (25)	14 (14)	58 (57)
Female	12 (12)	24 (24)	8 (8)	44 (43)
Average age, mean, (SD)	16.4 (1.4)	16.7 (1.3)	16.5 (1.4)	16.6 (1.4)
Spasticity, *n*. (%)	14 (14)	42 (43)	20 (20)	76 (75)
Hemiplegia, *n*. (%)	1 (1)	5 (5)	4 (4)	10 (10)
Diplegia, *n*. (%)	4 (4)	8 (8)	3 (3)	15 (15)
Tri/quadriplegia, *n*. (%)	9 (9)	29 (29)	13 (13)	51 (50)
Dystonia	4 (4)	3 (3)	5 (5)	12 (12)
Scoliosis	6 (6)	19 (19)	15 (15)	40 (40)
Gastrostomy	3 (3)	12 (12)	13 (13)	28 (28)
Autism spectrum disorders, *n* (%)	9 (9)	12 (12)	7 (7)	28 (28)
Psychotropic medication, *n*. (%)	13 (13)	9 (9)	6 (6)	28 (28)
Intellectual disability: moderate *n* (%)	7 (7)	6 (6)	0 (0)	13 (13)
Intellectual disability: severe *n*. (%)	12 (12)	10 (10)	4 (4)	26 (26)
Intellectual disability: profound	11 (11)	33 (33)	18 (18)	63 (61)
Ante-natal causes, *n*. (%)	22 (22)	33 (33)	8 (8)	63 (62)
Perinatal causes, *n*. (%)	8 (8)	12 (12)	7 (7)	27 (26)
Postnatal causes, *n*. (%)	1 (1)	3 (3)	8 (8)	12 (12)

**Table 2 children-09-01918-t002:** Contingency table and Fisher’s exact test comparing subjects with and without epilepsy. Gross Motor Function Classification System (GMFCS), Manual Ability Classification System [MACS], Communication Function Classification System (CFCS), Support Intensity Scale (SIS), Medical and Behavioral and Intellectual Disability (ID), Eating and Drinking Ability Classification System (EDACS).

Independent Variables		Presence of Epilepsy		Fisher’s Exact Test *p* Value Equals	Z Statistic	Odds Ratio	95% Confidence Intervals
		Yes	No				
MACS ≤ 4 versus MACS 5	Yes	45	3	<0.001	4.13	15.00	4.15 to 54.20
	No	27	27				
Neonatal epilepsy	Yes	54	0	<0.001	3.57	Infinity	Infinity
	No	18	30				
GMFCS ≤ 4 versus GMFCS 5	Yes	44	7	<0.001	3.31	5.16	1.95 to 13.61
	No	28	23				
EDACS ≤ 4 versus EDACS 5	Yes	25	4	0.035	2.09	3.45	1.08 to 11.01
	No	47	26				
CFCS ≤ 4 versus CFCS 5	Yes	41	10	0.049	2.14	2.64	1.08 to 6.44
	No	31	20				
Psychotropic medication	Yes	15	18	<0.001	3.68	0.17	0.06 to 0.44
	No	57	12				
Presence of spasticity	Yes	62	14	<0.001	3.91	7.08	2.65 to 18.88
	No	10	16				
SIS MED ≤ 10 versus SIS MED > 11	Yes	26	2	0.002	2.68	7.91	1.74 to 35.92
	No	46	28				
SIS behavioral ≤ 7 Versus SIS B. > 8	Yes	43	11	0.049	2.09	2.56	1.06 to 6.17
	No	29	19				
Walking capacity	Yes	24	21	<0.001	3.27	4.66	1.85 to 11.73
	No	48	9				
Surgeons	Yes	42	10	0.029	2.26	2.80	1.14 to 6.83
	No	30	20				
Truncal tone disorders	Yes	45	9	0.004	2.90	3.88	1.55 to 9.71
	No	27	21				
Gastrostomy	Yes	25	3	0.013	2.38	4.78	1.32 to 17.35
	No	47	27				
Neuromuscular scoliosis	Yes	34	6	0.013	2.48	3.57	1.30 to 9.79
	No	38	24				
Presence of feeding disorders	Yes	27	4	0.018	2.30	3.90	1.22 to 12.38
	No	42	26				
Presence of ID 4 versus ID ≤ 3	Yes	61	0	<0.001	3.96	326.2	18.59 to 5722
	No	11	30				

**Table 3 children-09-01918-t003:** Logistic multi-regression. The dependent variable was epilepsy yes/no.

LOGISTIC REGRESSION					
Variates	Odds Ratio		Standard Error	Z Ratio	Prob (>|z|) *p* Value
	Logar.	Linear			
Autism spectrum disorders (A)	−0.201	0.81	0.54	−0.369	0.712
Etiology (ET)	0.899	2.45	0.43	2.092	0.036
Sex (SE)	−0.45	0.63	0.50	−0.897	0.369
Scoliosis (NS)	1.08	2.95	0.54	1.982	0.047
CFCS	0.78	2.19	0.33	2.372	0.017
GMFCS	0.67	1.96	0.20	3.300	<0.001
Intellectual disability levels (ID)	0.93	2.54	0.34	2.698	0.006
Type of spasticity (SP)	0.61	1.85	0.20	3.048	0.002
EDACS	0.50	1.65	0.16	3.045	0.002

## Data Availability

Not applicable.

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
