# Peer review of "Prediction Model for Identifying Factors Associated with Epilepsy in Children with Cerebral Palsy"

_children, 2022, doi:10.3390/children9121918_

Round 1
Reviewer 1 Report
Children (ISSN 2227-9067)
The authors aimed to identify factors associated with epilepsy in children with CP using a prediction learning model. This could have prognostic interest, leading to a better tailor intervention in order to prevent epileptic crisis.
Pag 2, line 51 - Previous studies 51 had small sample sizes. The authors could include the reference of Peet work, since add information on the thematic with a sample of reasonable dimension. (1)
Page 2, line 91 -. Narrative notes were coded and entered an electronic database. Authors should assure in text that the very diversify academic formation of the researchers involved represented no bias on this step of the research.
Page 3, line 125 - Intractable epilepsy was defined as continued seizures despite adequate trials of at least two appropriate antiepileptic agents. Please clarify if in this sample, there was no other therapeutical intervention, such as ketogenic diet. (2)
Page 4, line 157, Table 2. Please create a footer with all the abbreviations present in the table.
Page 8, line 281 – Authors confirm the link between motor deficiency and the extent of spasticity with epilepsy. Then refer to feeding disabilities, namely needing gastrostomy. Both these parameters are strongly related, and this should be mentioned in text. (3)
Page 8, line 309 – Authors highlight the importance of early detection and implementation of specific interventions to improve prognosis and propose early specific interventions for each the morbidities previous mention (line 296). In this topic, refer that early dietary follow-up can be set up and will allow checking if food intake is adequate. Authors do not mention, but it’s very important performed regular anthropometric assessment. Even with challenges on the assessment to obtain simple measures like weight and height in some children with CP, there are alternative methods to assess nutritional status, as presented in Pinto recent publish work that must be highlighted. (4)
1 - Haridas B, Kossoff EH. Dietary Treatments for Epilepsy. Neurol Clin. 2022 Nov;40(4):785-797. doi: 10.1016/j.ncl.2022.03.009.
2- Pietrzak D, Kasperek K, Rękawek P, Piątkowska-Chmiel I. The Therapeutic Role of Ketogenic Diet in Neurological Disorders. Nutrients. 2022 May 6;14(9):1952. doi: 10.3390/nu14091952. PMID: 35565918; PMCID: PMC9102882.
3 - Dahlseng MO, Andersen GL, DA Graca Andrada M, Arnaud C, Balu R, De la Cruz J, et al; Surveillance of Cerebral Palsy in Europe Network. Gastrostomy tube feeding of children with cerebral palsy: variation across six European countries. Dev Med Child Neurol. 2012 Oct;54(10):938-44. doi: 10.1111/j.1469-8749.2012.04391.x.
4 - Pinto C, Borrego R, Eiró-Gomes M, Casimiro I, Raposo A, Folha T, et al. Embracing the Nutritional Assessment in Cerebral Palsy: A Toolkit for Healthcare Professionals for Daily Practice. Nutrients. 2022 Mar 11;14(6):1180. doi: 10.3390/nu14061180.
Author Response
The authors aimed to identify factors associated with epilepsy in children with CP using a prediction learning model. This could have prognostic interest, leading to a better tailor intervention in order to prevent epileptic crisis.
Reply: Thank you for reviewing our manuscript and providing suggestions for improving it.
Reviewer 1
Pag 2, line 51 - Previous studies had small or limited sample sizes. The authors could include the reference of Peet work, since add information on the thematic with a sample of reasonable dimension. (1)
Reply: We included the reference: Haridas B, Kossoff EH. Dietary Treatments for Epilepsy. Neurol Clin. 2022 Nov;40(4):785-797. doi: 10.1016/j.ncl.2022.03.009.
Page 2, line 91 -. Narrative notes were coded and entered an electronic database. Authors should assure in text that the very diversify academic formation of the researchers involved represented no bias on this step of the research.
Reply: We added the text as follow:
Page 3, line 110: To minimize the biases regarding the diversified academic background of the researchers, only members working together for at least 10 years were included in the study and the results were discussed periodically under the supervision of a child epileptic senior spe-cialist.
Page 3, line 125 - Intractable epilepsy was defined as continued seizures despite adequate trials of at least two appropriate antiepileptic agents. Please clarify if in this sample, there was no other therapeutical intervention, such as ketogenic diet. (2)
Reply: We added the text as follow:
Page 4, line 159: In our cohort, there was no other therapeutical intervention, such as ketogenic diet
Page 4, line 157, Table 2. Please create a footer with all the abbreviations present in the table.
Reply: We added the text as follow:
Page 8, line 288, Table 2. Manual Ability Classification System [MACS], Gross Motor Function Classification Sys-tem (GMFCS), Eating and Drinking Ability Classification System (EDACS), Communication Function Classification System (CFCS), Support intensity Scale (SIS) Medical and Be-havioral and Intellectual Disability (ID).
Page 8, line 281 – Authors confirm the link between motor deficiency and the extent of spasticity with epilepsy. Then refer to feeding disabilities, namely needing gastrostomy. Both these parameters are strongly related, and this should be mentioned in text. (3)
Reply: We added the text as follow:
Page 10, line 339: As highlighted by Dahlseng et al. [23], motor deficiency, the extent of spasticity, feeding disabilities, namely needing gastrostomy, and epilepsy are strongly related. Artificial feeding should be proposed after multidisciplinary concertation to avoid deficiencies (vitamins and trace elements) that facilitate the occurrence of epileptic seizures in an epileptic patient. Dysphagia or digestive troubles must be considered in an epileptic patient who must swallow antiepileptic drugs regularly every day. Also, the decision to feed a child with a gastrostomy needs to be made appropriately after a multidisciplinary assessment. Caregivers and parents have to be included, considering acceptance also de-pends on cultural parameters.
Dahlseng MO, Andersen GL, DA Graca Andrada M, Arnaud C, Balu R, De la Cruz J, et al; Surveillance of Cerebral Palsy in Europe Network. Gastrostomy tube feeding of children with cerebral palsy: variation across six European countries. Dev Med Child Neurol. 2012 Oct;54(10):938-44. doi: 10.1111/j.1469-8749.2012.04391.x.
Page 8, line 309 – Authors highlight the importance of early detection and implementation of specific interventions to improve prognosis and propose early specific interventions for each the morbidities previous mention (line 296). In this topic, refer that early dietary follow-up can be set up and will allow checking if food intake is adequate. Authors do not mention, but it’s very important performed regular anthropometric assessment. Even with challenges on the assessment to obtain simple measures like weight and height in some children with CP, there are alternative methods to assess nutritional status, as presented in Pinto recent publish work that must be highlighted. (4)
Reply: We added the text as follow
Page 11 line 385: Early dietary follow-up can be set up and will allow checking if food intake is adequate. Performing regular anthropometric assessments is very important. Even with challenges in the evaluation to obtain simple measures like weight and height in some children with CP, there are alternative methods to assess nutritional status. A toolkit was recently created to help caregivers to detect malnutrition in children with cerebral palsy (29)
Pinto C, Borrego R, Eiró-Gomes M, Casimiro I, Raposo A, Folha T, et al. Embracing the Nutritional Assessment in Cerebral Palsy: A Toolkit for Healthcare Professionals for Daily Practice. Nutrients. 2022 Mar 11;14(6):1180. doi: 10.3390/nu14061180.
Reviewer 2 Report
Dr. Carlo M. Bertoncelli and his colleagues implemented a prediction model, Epi-Predict-Med, to forecast neuromuscular scoliosis and feeding disorders needing gastrostomy , factors associated with intellectual disabilities, autism spectrum disorder, and neuromuscular hip dysplasia as epilepsy associated factors , in patients with (cerebral palsy) CP. They have analyzed the data collected from the existing records in the databases (2005 to 2020) in 102 children with cerebral palsy meeting the inclusion criteria)
This study is very interesting, it addresses a knowledge gap in the literature given the scarcity in this respect however, I have some serious concerns that hinder the publication in the current form and need to be addressed by the authors to improve the manuscript.
1. Introduction and discussions
Minor
1. Lines 37-38
In addition to [1], the authors should succinctly present the pro-inflammatory processes of epilepsy and cerebral palsy for the reader to understand the background leading to the brain injury and eventually to epileptic events and CP during the early stages of brain development. For example, you could use Neamtu et al. paper A Decision-Tree Approach to Assist in Forecasting the Outcomes of the Neonatal Brain Injury Int. J. Environ. Res. Public Health 2021, 18(9), 4807; https://doi.org/10.3390/ijerph18094807, or any other manuscript in this respect:
„ At a molecular level, many pathways were incriminated in apoptosis of premyelinating oligodendrocytes or subplate neurons involved in perinatal brain development. Glutamate rising concentrations or free radical reactive species (both oxygen and hydrogen) in hypoxic-ischemic encephalopathy, inflammatory cytokines such as TNF-α, IL-1b, IL-6, 12, 15, 18 from activated microglia and astrocytes, and low pH in infections, free iron secondary to cerebral hemorrhage were extensively mentioned in both white and grey matter injuries as important triggers for epileptic events.”
2. Lines [40-42]
„ The most frequent types of seizures are complex focal and secondary generalized seizures, and early evaluation is strongly recommended [2]” as children with cerebral palsy show the tendency of having an earlier epilepsy onset, and the degree of severity is positively correlated with the CP’s severity. I recommend adding this clarification.
2. Materials and methods
Major
1. What exactly do the authors mean when they refer to ...” two sites (65 Hospital, 37 Day Hospital), 102 children met the inclusion criteria”.
2. Tables 1,2 display results, therefore should be presented in the results section, not in the methodology.
3. Lines 152-156 regarding the initial statistical analysis should mention the reference p-value or alpha level (most probably 0.05). Moreover, for the logistic regression it should be outlined that they have used a relaxed p-value < 0.2 and references from the literature should be mentioned in this respect(lines 162-165).
4. The authors should provide an appendix with a table containing the AUROC values of the independent variables from the selected tuple.
3. Discussions
Minor
1. Lines 340-341“ Maybe an algorithm including these parameters could help clinicians anticipate diagnosis and take the most appropriate therapeutic decision?” This would sound better as a statement not as a question.
Major
1. The authors need to compare their accuracy and the average score and high sensitivity with other reports in the literature (lines 342-345)
1. Limitations – the lack of references in the literature regarding the associated factors is the critical point supporting their approach in terms of originality and knowledge gap. Hence it shouldn’t be presented in this way as a limitation. The main limitation indeed is related to the small of patients included in the study with and most importantly to the retrospective nature of the study with the possibility of residual confounding. Therefore, their model performance should be interpreted with caution given these circumstances, and the authors should mention this aspect in the limitations section. Future multicentric research studies should employ randomized control trials on larger cohorts to address these issues.
Author Response
Comments and Suggestions for Authors
Dr. Carlo M. Bertoncelli and his colleagues implemented a prediction model, Epi-Predict-Med, to forecast neuromuscular scoliosis and feeding disorders needing gastrostomy , factors associated with intellectual disabilities, autism spectrum disorder, and neuromuscular hip dysplasia as epilepsy associated factors , in patients with (cerebral palsy) CP. They have analyzed the data collected from the existing records in the databases (2005 to 2020) in 102 children with cerebral palsy meeting the inclusion criteria)
This study is very interesting, it addresses a knowledge gap in the literature given the scarcity in this respect however, I have some serious concerns that hinder the publication in the current form and need to be addressed by the authors to improve the manuscript.
Reply: Thank you for reviewing our manuscript and suggestions for improving it.
- Introduction and discussions
Minor
- Lines 37-38
In addition to [1], the authors should succinctly present the pro-inflammatory processes of epilepsy and cerebral palsy for the reader to understand the background leading to the brain injury and eventually to epileptic events and CP during the early stages of brain development. For example, you could use Neamtu et al. paper A Decision-Tree Approach to Assist in Forecasting the Outcomes of the Neonatal Brain Injury Int. J. Environ. Res. Public Health 2021, 18(9), 4807; https://doi.org/10.3390/ijerph18094807, or any other manuscript in this respect:
„ At a molecular level, many pathways were incriminated in apoptosis of premyelinating oligodendrocytes or subplate neurons involved in perinatal brain development. Glutamate rising concentrations or free radical reactive species (both oxygen and hydrogen) in hypoxic-ischemic encephalopathy, inflammatory cytokines such as TNF-α, IL-1b, IL-6, 12, 15, 18 from activated microglia and astrocytes, and low pH in infections, free iron secondary to cerebral hemorrhage were extensively mentioned in both white and grey matter injuries as important triggers for epileptic events.”
Reply: We added the text as follow
Page 2 line 60: At a molecular level, many pathways were incriminated in apoptosis of premyelinating oligodendrocytes or subplate neurons involved in perinatal brain development. Glutamate rising concentrations or free radical reactive species (both oxygen and hydrogen) in hypoxic-ischemic encephalopathy, inflammatory cytokines such as TNF-α, IL-1b, IL-6, 12, 15, 18 from activated microglia and astrocytes, and low pH in infections, free iron secondary to cerebral hemorrhage were extensively mentioned in both white and grey matter injuries as important triggers for epileptic events.
- Lines [40-42]
„ The most frequent types of seizures are complex focal and secondary generalized seizures, and early evaluation is strongly recommended [2]” as children with cerebral palsy show the tendency of having an earlier epilepsy onset, and the degree of severity is positively correlated with the CP’s severity. I recommend adding this clarification.
Reply: We added the text as follow
Page 2 line 49: as children with CP tend to have an earlier epilepsy onset, and the degree of severity is positively correlated with the CP's severity.
- Materials and methods
Major
- What exactly do the authors mean when they refer to ...” two sites (65 Hospital, 37 Day Hospital), 102 children met the inclusion criteria”.
Reply: We clarified and modified the text as follow
Page 2, line 95: 102 children (65 hospitalized, 37 in day hospital), met the following inclusion criteria.
- Tables 1,2 display results, therefore should be presented in the results section, not in the methodology.
Reply: We moved the Tables 1 and 2 in the results section
- Lines 152-156 regarding the initial statistical analysis should mention the reference p-value or alpha level (most probably 0.05).
Reply: We added the sentence:
Page 5, line 186: with the referenced p-value of 0.05.
Moreover, for the logistic regression it should be outlined that they have used a relaxed p-value < 0.2 and references from the literature should be mentioned in this respect (lines 162-165).
Reply: We add the reference:
Solla F, Tran A, Bertoncelli D, Musoff C, Bertoncelli CM. Why a P-Value is Not Enough. Clin Spine Surg. 2018 Nov;31(9):385-388. doi: 10.1097/BSD.0000000000000695. PMID: 30036210.
- 4. The authors should provide an appendix with a table containing the AUROC values of the independent variables from the selected tuple.
Reply: We recognize that with a limited cohort of patients (102) the AUROC values would not be indicative. Indeed, we have in plan a new research on a greater number of patients (265). We will study the ROC curves and AUROC values on this new bigger dataset in order to validate the results of our research.
We added this in the “limitation of the study “
Page 12, line 427: The main limitations of our study were the relatively small number of patients included that can lead to an approximate evaluation of the predictive performance of our algorithm
- Discussions
Minor
- Lines 340-341 Maybean algorithm including these parameters could help clinicians anticipate diagnosis and take the most appropriate therapeutic decision?” This would sound better as a statement not as a question.
Reply: We modified the sentence:
Page 12 line 413: An algorithm that includes these parameters could help clinicians anticipate multiple diagnoses and make the most appropriate treatment decision.
Major
- The authors need to compare their accuracy and the average score and high sensitivity with other reports in the literature (lines 342-345)
Reply: As we wrote in the Discussion paragraph, as far as we know a prediction model has never been implemented to assess the prevalence of epilepsy in children with CP. Apart our previous researches (6-8) applying the PredictMed model, we have not found any studies that report the accuracy, sensitivity and specificity. We then compared the present results with our previous results (6-8) and we added:
Page 10, line 320: Previous studies [8] stated that CP children with epilepsy were twice as likely to develop scoliosis and truncal tone disorders and undergo surgeries. Epi-PredictMed scored an average accuracy, sensitivity, and specificity of 82%, improving the previous results of 74% [8].
Page 11, line 348: We also found a strong association (odds ratio > 2) between epilepsy and profound intellectual and communication disabilities, as previously reported [20, 21, 24]. In addition, in the present study, the Epi-Predicted sensitivity of 98% allows for the more precise identification of children at risk of epilepsy. And this allows for the personalization of treatments.
Furthermore, we updated the bibliography and compared our results with a recent study by Archana and al. (Seizure, 2022) and we added:
Page 10 line 339: As highlighted by Dahlseng et al. [23], motor deficiency, the extent of spasticity, feeding disabilities, namely needing gastrostomy, and epilepsy are strongly related. Artificial feeding should be proposed after multidisciplinary concertation to avoid deficiencies (vitamins and trace elements) that facilitate the occurrence of epileptic seizures in an epi-leptic patient. Dysphagia or digestive troubles must be considered in an epileptic patient who must swallow antiepileptic drugs regularly every day. Also, the decision to feed a child with a gastrostomy needs to be made appropriately after a multidisciplinary as-sessment. Caregivers and parents have to be included, considering acceptance also de-pends on cultural parameters.
17.1 K A, Saini L, Gunasekaran PK, Singh P, Sahu JK, Sankhyan N, Sharma R, Bhati A, Yadav J, Sharawat IK. The Profile of Epilepsy and its characteristics in Children with Cerebral Palsy. Seizure. 2022 Oct;101:190-196. doi: 10.1016/j.seizure.2022.08.009. Epub 2022 Aug 24. PMID: 36070632.
- Limitations – the lack of references in the literature regarding the associated factors is the critical point supporting their approach in terms of originality and knowledge gap. Hence it shouldn’t be presented in this way as a limitation. The main limitation indeed is related to the small of patients included in the study with and most importantly to the retrospective nature of the study with the possibility of residual confounding. Therefore, their model performance should be interpreted with caution given these circumstances, and the authors should mention this aspect in the limitations section. Future multicentric research studies should employ randomized control trials on larger cohorts to address these issues.
Reply: We modified the sentence:
Page 12 line 427: The main limitations of our study were the relatively small number of patients included that can lead to an approximate evaluation of the predictive performance of our algorithm and the retrospective nature of the study with the possibility of residual con-founding. Therefore, the Epi-PredictMed model performance should be interpreted with caution. Therefore, we plan a multicentric research study employing randomized control trials on larger cohorts to address these issues
Round 2
Reviewer 2 Report
The authors have addressed my comments point-by-point. They have definitely and sufficiently improved the clarity of the manuscript in reaction to the remarks that have been made. Their design is interesting and shows promise for future research approaches. I recommend the manuscript to be published.